# Hand Grip Strength Relative to Waist Circumference as a Means to Identify Men and Women Possessing Intact Mobility in a Cohort of Older Adults with Type 2 Diabetes

**DOI:** 10.3390/biomedicines11020352

**Published:** 2023-01-26

**Authors:** Ofer S. Kis, Assaf Buch, Roy Eldor, Daniel S. Moran

**Affiliations:** 1The Department of Health Systems Management, Ariel University, Ariel 40700, Israel; 2Department of Nutritional Sciences, School of Health Sciences, Ariel University, Ariel 40700, Israel; 3Institute of Endocrinology, Metabolism and Hypertension, Tel Aviv Sourasky Medical Center, Tel-Aviv 40700, Israel

**Keywords:** hand grip strength, mobility-intact, older adults, type 2 diabetes mellitus

## Abstract

Possessing intact mobility in older adults assures their continued independence. The early identification of reduced mobility in older adults with type 2 diabetes (T2DM) is paramount for preventing their future physical deterioration. Hand grip strength (HGS), relative to body size, is associated with mobility in older T2DM patients. This study aims to identify an HGS index that best identifies mobilityintact older T2DM patients, along with its optimal cut-off point. The baseline data are from a cohort of 122 older T2DM patients (59% women) (mean age of 70.2 ± 4.4 years). Three mobility tests encompassing three main mobility domains were measured, including usual gait speed (UGS), timed up and go (TUG), and a two-minute walk test (2MWT). Passing scores were defined as those either above the established cut-off points or above the 25th percentile of population norms. Passing all three tests was considered as possessing intact mobility. Receiver operating characteristic (ROC) curves of the most relevant HGS indices were constructed to determine the area under the curve (AUC) that best identifies patients with intact mobility. In a sample of 122 older adults with T2DM, 63.9% of women and 60% of men were found to possess intact mobility. HGS relative to waist circumference (WC) was found to have the strongest association with intact mobility, presenting the highest AUC in both men (0.78) and women (0.72) for discriminating mobility status, with an optimal cut-off of 0.355 (kg/cm) and 0.245 (kg/cm) in men and women, respectively. HGS relative to WC best differentiated between mobility-intact older adults with T2DM and those with mobility limitations, especially in men. Using HGS/WC as a simple and safe screening mode for mobility in a clinical setting could potentially identify older patients with T2DM that require therapeutic interventions.

## 1. Introduction

Mobility, generally referred to as a person’s ability to change their position and move from one place to another by walking [1], is a crucial component of healthy aging [2] and continued independence [3]. It is generally determined as to whether older adults are “mobilityintact” through the use of questionnaires (“having no difficulty in walking”) [4], however, being “mobility-limited” is determined either by self-report surveys or by mobility test evaluations. Declining mobility affects ~30% of older adults (with a range of 22.5% to 46.7%) [5], which occurs at an individual trajectory [3]. The high rate of mobility decline requires the implementation of therapeutic interventions [6], which compel clinicians to routinely conduct physical performance testing to identify older adults at early stages of functional [7] and mobility decline [8]. The accelerated decline in mobility performance [9,10] exhibited by older adults with type 2 diabetes mellitus (T2DM) reaches a two–three-fold greater prevalence compared to healthy older adults [11]. Due to the accompanying increase in the physical disability rate in diabetic patients (12), the need arises to routinely evaluate their mobility status in primary care settings [12]. Some of the most frequently performed tests to evaluate mobility include usual gait speed (UGS) [13,14,15], timed up and go (TUG) [16,17,18], and long corridor walks (two-minute, six-minute, and 400 m walk tests) [5,19,20], which predict future mobility limitations even in populations reporting no walking difficulties [21]. As no single test encompasses all aspects of mobility [1], it stands to reason that performing multiple mobility tests could provide a more comprehensive assessment of mobility status [21], especially in older T2DM patients who are prone to fatigue and instability during walking [9]. Despite their prognostic value [22,23] and relative ease of administration [24], such tests are seldomly performed in clinical practice concerning diabetes [25]. 

In older adults, muscle strength is a discriminator of both functional abilities [26] and mobility disability [27]. Muscle strength is predominantly evaluated by handgrip strength (HGS) testing [28] due to its association with functional limitations and its ease of administration [29]. HGS is markedly influenced by gender, age, body size [12,30,31], the presence of T2DM [12,30], and especially obesity [32,33,34,35,36], albeit generally normalized to either body mass index (BMI) [37,38,39] or body weight [40,41,42].

Previous studies have been performed to find clinically relevant criteria that identify HGS cut points (absolute or relative to body size), which identify muscle weakness associated with mobility impairment, most frequently defined as a slow UGS [43,44,45,46,47]. Finding an HGS index to be used as a proxy for mobility limitation testing in a primary care setting is of great importance [12,48], especially where consultation time is of the essence [49,50,51,52].

To the best of our knowledge, this is the first investigation attempting to explore the use of a hand grip index that can best identify mobility-intact older adults with T2DM. The second goal of this study is to identify the most relevant cut-off points that best predict intact mobility in this population. 

## 2. Materials and Methods

### 2.1. Sample and Procedure

The present investigation is an analysis of baseline measurements (cross-sectional) of a clinical trial conducted in the endocrinology unit at the Tel-Aviv Medical Center. The trial had two phases—a pilot study evaluating the physical characteristics of a sample of older adults with diabetes and/or metabolic syndrome, and a randomized clinical trial—the CEV-65 trial [53]. The present cohort was based on the following: (a) T2DM patients enrolled in the pilot study (10 men and 12 women) and (b) T2DM patients enrolled in the CEV-65 trial (40 males and 60 females), which investigated the efficacy of resistance strength training, pharmacotherapy (empagliflozin), and a diet intervention (vegetarian/Mediterranean) on the prevention of sarcopenia and/or frailty measures [53]. The subjects belonged to outpatient clinics in the medical center (e.g., endocrinology, osteoporosis, diabetes, and metabolic syndrome), or were patients of resident physicians from community clinics located in the Tel Aviv area. The study (both phases) has received ethics approval from the Tel-Aviv Sourasky Medical Center Institutional Review Board(Approval code: MOH002218_2018-03-21_; Approval date: 20 March 2018). All of the participants provided written informed consent to participate in the study.

### 2.2. Eligibility Criteria

Overall, 122 older (≥65 years) male and female (59% female) T2DM patients, diagnosed in accordance with the American Diabetes Association guidelines [54], were recruited. The main inclusion criteria included the following: performing any leisure aerobic physical activity ≤2 days a week, walking independently either with or without an assistance device, and HbA1C of ≥6.5% to ≤8%. The exclusion criteria included the following: performing any resistance training within the past six months, the use of any steroids, severe peripheral neuropathy, end-stage renal failure, a history of stroke, myopathy, motor functional disorders, and treatment with SGLT-2 inhibitors. The recruitment process, as well as the complete eligibility criteria, are described in detail elsewhere [53].

### 2.3. Exposure Assessment 

#### 2.3.1. Hand Grip Strength (HGS) Assessment

HGS was assessed by a handheld dynamometer (Jamar® Sammons Preston Rolyan, Chicago, IL, USA). Subjects were seated in an upright position with their arm along their side, the elbow bent at 90°, the arm supported horizontally by a tester, and their feet firmly planted on the floor. The width of the dynamometer handle was adjusted to fit the size of the hand (generally using the second smallest grip). One orientation trial was performed for each hand before three trials, alternating between arms, were performed. The highest of the six measurements was recorded [55]. 

#### 2.3.2. Anthropometry

Body weight as well as body composition measurements were obtained through the direct segmental multifrequency bioelectrical impendence analysis (BIA) technique method (InBody 770 body composition analyzer, InBody Co., Ltd, Seoul, Korea), which was shown to be accurate and validated against DXA [56] in our cohort of older adults with diabetes [57]. Measurements were performed by the same technician throughout the study and included skeletal muscle mass, fat mass, and % body fat. Height was measured electronically to the nearest 0.1 cm, and body mass index (BMI) was calculated. Waist circumference (WC) was measured twice (the average was calculated) using a designated tape measure. Obesity was defined either by a BMI > 30 [58], WC (men: >102 c” m; women: >88 c” m) [59], or BF% (men: ≥32.3; women: ≥44.1) [60]. 

HGS, in relation to various anthropometric measurements, was computed with HGS as the numerator and anthropometric measurements as the denominator [61], yielding relative HGS indices. The following HGS indices were computed: (a) HGS (absolute value); (b) HGS/body weight (HGS/BW); (c) HGS/body mass index (HGS/BMI); (d) HGS/waist circumference (HGS/WC); (e) HGS/height (HGS/H); (f) HGS/body fat percent (HGS/BFP); (g) HGS/body fat mass (HGS/BFM); and (h) HGS/skeletal muscle mass (HGS/SMM). 

### 2.4. Outcome Assessment 

#### 2.4.1. Usual Gait Speed (UGS)

Patients were asked to walk a 4 m distance from a static start at their regular walking pace. Timing with a stopwatch began when the first foot passed the starting line and ended when the first foot passed the finish line [28]. Patients performed one orientation trial before performing two tests, and the averaged time was computed. For comparison with the NIH scores, individual 4 m completion times were averaged, divided by 4 (meters), and then expressed in meters/second (m/s). The cut-off point for passing the UGS test was considered to be scoring equal or above the 25th percentile of the NIH 4-meter walk gait speed test: 1.00 m/s and 0.91 m/s or below (ages of 60–69 y) as well as 0.88 m/s and 0.84 m/s or below (ages of 70–85 y) in men and women, respectively [62]. 

#### 2.4.2. Timed Up and Go Test (TUG)

Patients were seated in an armchair with a seat height of ~44 cm, three meters away from a cone. Subjects were timed on their ability to stand up from the chair, walk the 3 m course, turn safely around the cone, walk back, and sit down again (in seconds). The patients performed one preliminary trial before an actual test. The cut-off point for passing the TUG test for both genders was considered to be a time of <12 s [18]. 

#### 2.4.3. Two-Minute Walk Test (2MWT)

In order to reduce patients’ fatigue during long corridor walks, the 2MWT was performed as it is highly correlated to the common six-minute walk test [63]. Patients were asked to walk as quickly and safely as possible in a spacious and quiet hospital corridor between two cones set 15.24 m apart. During the test, the elapsed time was called out 1 min into the test and also at 1:40 s. The total distance was computed by multiplying the number of single-side completions by 15.24 (50 ft) and adding the marked distance where the 2 min time was terminated [64]. The cut-off points for passing the 2MWT were considered to be a score above the 25th percentile (stratified by age and gender) of the “NIH Toolbox 2 Minute Walk Endurance Test, 2012”. In order to compare individual patient scores with NIH tables, scores were converted from meters into feet. 

#### 2.4.4. Mobility Status 

The level of mobility limitations is generally categorized based on self-reported measures [5]. Despite their advantages, self-reported measures have been shown to overestimate mobility performance compared to performance-based measures [65], which: (a) provide objective results; (b) are sensitive to change; and (c) reflect specific functional subdomains [66]. To sufficiently assess a patient’s performance, there is the need to evaluate a patient’s deficits across multiple domains. The results of multiple outcome measures, along with their disparate scales, might increase the difficulty in reaching an integrated assessment, calling for the creation of a single composite measure [67]. 

To create a single tool that comprises the multiple constructs imbedded in our three mobility tests, we created a mobility composite score [68] composed of a dichotomous score of 0 and 1, where a score of 0 was given for a test score below an established cut-off point and a score of 1 was given for tests surpassing a cut-off point. Adding individual test scores produced a single composite score [69]. A patient that received a score of 3 out of 3 (passing all three mobility tests) was considered as being “mobility-intact”, while those that failed to pass cut-off points on any of the mobility tests, were considered to have mobility limitations. 

### 2.5. Statistical Analysis 

Descriptive statistics were used for the clinical and mobility data. For all of the nominal variables absolute frequencies and percentages were calculated. For all of the continuous variables means and standard deviations were calculated. Multiple univariate independent t-tests were performed between each HGS index and passing the cut-off points of each mobility test as well as all 3 tests. In order to examine the dependency between clinical data (dichotomous variables) and passing the cut-off points of each mobility test as well as all 3 tests, a chi-square test with Yates correction or a Fisher exact test was performed. Receiver operating characteristic (ROC) curve analysis, including area under the curve (AUC) and accuracy, was performed in order to find the optimum thresholds for HGS indices. Odds ratios (ORs) with a 95% confidence interval were calculated in order to test the optimum threshold prediction of passing each mobility test and all 3 tests. Data were prepared in Microsoft Excel, and statistical analyses were conducted using SPSS statistical software (Version 21) IBM Corp, Armonk, NY, USA. The criterion for significance was alpha (α) = 0.05 (two-sided).

## 3. Results

### 3.1. Participants’ Characteristics

#### 3.1.1. Clinical Characteristics

Older patients with T2DM (122: 50 men and 72 women) are listed in Table 1. The mean age was 70.3 ± 4.4 years, the mean HbA1c was 7.4 ± 1.14, and the mean BMI was 32 ± 5.95 kg/m^2^. The prevalence of obesity by body mass index (BMI > 30 kg/m^2^), WC (men: >102 cm; women: >88 cm), and body fat percentage (men: >32.3%; women: >44.1%) was 64%, 92%, and 57% in men and 64%, 96%, and 70% in women. The mean diabetes duration was 12.8 ± 10.2 years in men and 14.2 ± 8.7 years in women. Low Vit D levels (<25 mg/mL) were found in 31% of patients. Diabetes complications (neuropathy, retinopathy, chronic kidney disease, and ischemic heart disease) were present in 39% of the cohort. The mean absolute HGS was 40.4 ± 6.6 kg and 24.9 ± 4.7 kg in men and women, respectively. HGS/BW and HGS/BMI were 0.44 ± 0.1 and 1.30 ± 0.31 as well as 0.32 ± 0.08 and 0.80 ± 0.21 in men and women, respectively. Mean HGS/WC and HGS/PBF were 0.36 ± 0.08 and 1.21 ± 0.38 as well as 0.23 ± 0.05 and 0.58 ± 0.14 in men and women, respectively. HGS/SMM was 1.12 ± 0.28 in men and 0.99 ± 0.28 in women. All HGS indices were significantly higher in men compared to women (*p* < 0.001). 

#### 3.1.2. Mobility Characteristics

In our cohort of older adults with T2DM, 60% of men and 63.9% of women were found to possess intact mobility by passing the cut-off points of all of the mobility tests. An analysis of individual tests indicated that the largest number of patients passed the UGS test (87% of men and 78% of women), with 82% of men and 77.8% of women passing the TUG test. The 2MWT yielded the lowest rate of patients passing the cut-off points (71.4% of men and 75% of women). No significant differences were found between genders in the percentage of patients passing either individual or all three tests. In men, 20% failed a single mobility test, while in women, 20.9 % failed a single test (Table 2). 

### 3.2. Determination of Optimal HGS Indices for Analysis Using Univariate Analysis

Multiple univariate independent t-tests, performed between each HGS index and patients passing the cut-off points of all three tests (Appendix A), revealed three HGS indices possessing the highest associations with passing all of the mobility tests. These HGS indices were shared by both men and women: HGS/WC, HGS/BMI, and HGS/W. HGS/WC possessed the highest values of the three HGS indices in men, t (48) = 3.548, *p* = 0.001, and women, *t* (70) = 3.255, *p* = 0.002. In women, the UGS test was not associated with any HGS indices. 

### 3.3. Identification of Confounders

Chi-square tests with Yates correction were performed between each possible confounder (as a dichotomous variable) and passing individual as well as all three mobility tests (Appendix A). Confounders included the following: (a) Being physically active (performing any leisure aerobic physical activity). (b) Diabetes complication (diagnosis of neuropathy, retinopathy, chronic kidney disease, or ischemic heart disease). (c) BMI (≥30). (d) Vit D (<25 ng/mL). (e) Polypharmacy (≥eight drugs). (f) Diabetes duration (≥10 years). An analysis revealed that being physically active was found to be associated with passing all of the mobility tests in both men (*p* = 0.021) and women (*p* = 0.008).

### 3.4. Optimum Thresholds for HGS Indices Using Receiver Operating Characteristic (ROC) Curve Analysis and Testing Their Prediction for Intact Mobility Using Odds Ratios (ORs)

The HGS index with the highest AUC that best predicted passing all three mobility tests (being “mobility-intact”) was found to be HGS/WC in both men and women. The HGS/WC AUCs, along with their corresponding optimal cut-off points for identifying mobilityintact older T2DM patients and individual mobility tests, are presented in Table 3. The area under the curve (AUC) of mobility-intact men was 0.781 (95% CI 0.649–0.913), with an optimal HGS/WC cut-off point of 0.355 (kg/cm), a sensitivity of 73.3%, and a specificity of 80.0%. The AUC of mobility-intact women was 0.725, (95% CI 0.609–0.842), with an optimal HGS/WC cut-off point of 0.245 (kg/cm), a sensitivity of 54.3%, and a specificity of 84.6% (Figure 1).

An analysis of individual mobility tests revealed that, in men, the highest AUC was in the TUG test, at 0.827 (95% CI 0.704–0.950), with an optimal HGS/WC cut-off point of 0.355 (kg/cm), a sensitivity of 63.4%, and a specificity of 100%. In women, the highest AUC was in the 2MWT, of 0.751 (95% CI 0.623–0.880), with an optimal HGS/WC cut-off point of 0.215 (kg/cm), a sensitivity of 63.4%, and a specificity of 100%.

The odds of passing all three mobility tests using the cut-off points of HGS/WC were significantly higher compared to patients below cut-off points (Table 4) (men OR = 8.2, 95% CI 1.3–30.1) (women OR = 3.0, 95% CI 1.1–8.1). The 2MWT had the highest odds of passing the test using HGS/WC cut-off points (men OR = 13.1, 95% CI 2.5–68.7) (women OR = 5.6, 95% CI 1.7–18.4). 

## 4. Discussion

The primary and novel finding of our study revealed that HGS/WC has a moderate predictive ability to distinguish between mobility-intact older men with T2DM and those possessing mobility limitations. In women, the predictive power of HGS/WC was somewhat lower, yet still acceptable. It stands to reason that in a cohort of primarily obese T2DM older adults, measures of obesity [33,70], and in particular WC, would have such a strong and negative association with both HGS relative to WC [33,38,71] and mobility [72,73,74]. This finding is especially consequential since WC is consistently used to determine risks associated with central adiposity [75], and is both easily obtained as well as recommended to be routinely measured in clinical practice [76]. 

Searching for an HGS index cut point that can best distinguish between mobility-intact older adults and those at risk of mobility limitations has been the objective of several investigations. Choquette et al. [61] evaluated the following HGS indices, (expressing ratios of muscle strength and adiposity) with respect to increasing the risks of having mobility limitations: HGS/BW, HGS/BMI, HGS/fat mass, and HGS/fat mass index (kg/m^2^). Mobility limitations were considered as being in the lowest quartile on a mobility score composed of five tests of lower extremity functions: TUG, chair stand, UGS, fast-paced gait speed, and one-leg stand. HGS/BMI was selected for further analyses based on its partial correlation with mobility score (*r* = 0.32, *p* < 0.001), and revealed that the odds of having mobility limitations were 4.4 times higher for subjects that possessed the lowest HGS/BMI scores. Choquette’s subjects possessed lower obesity scores compared to our subjects (BMI 28 vs. 32 kg/kg/m^2^), especially with respect to women’s BW (66 vs. 80 kg), which greatly affects muscle strength. Choquette concluded that HGS indices are convenient to use in clinical practice and that muscle strength as well as adiposity may be more important in the prediction of mobility limitations than either variable by itself [61]. Dong et al. [77] examined several HGS indices, including HGS/BW, HGS/BMI, HGS/PBF, HGS/FM, HGS/fat-free mass, HGS/LBM, HGS/SMM, and HGS/upper limb muscle mass. Mobility limitations were defined as scoring in the lowest 20% of outcomes in either the TUG or the UGS (4 m walk). Compared to our individual UGS or TUG test results, the prevalence of subjects possessing mobility limitations was somewhat higher in Dong’s study (24% vs. 22% in men and 27% vs. 22% in women, respectively); however, compared to our criteria for mobility limitations (performing below cut-off points on any mobility test), our prevalence rate was much higher (40% and 36% in men and women, respectively). Dong et al. identified HS/body fat mass in men (AUC = 0.723, a sensitivity of 86.4%, and a specificity of 44.6%) and HS/weight for women (AUC = 0.684, a sensitivity of 53.9%, and a specificity of 72.1%) as the HGS indices that best distinguish between mobility statuses. The mean BMI in Dong’s study was much lower compared to our study (25 kg/kg/m^2^), with 12.5% of his subjects being T2DM patients [77]. It should be emphasized that in both of the above studies mobility cut-off points were determined by the results of the studies’ subjects, compared to cut-offs based on either established cut-offs or percentiles of the general population, as implemented in our investigation. 

The Sarcopenia Definitions and Outcomes Consortium (SDOC) performed several investigations to develop diagnostic cut points for muscle strength that identify older adults at increased risk of mobility limitations (defined as UGS < 0.8 m/s) [27]. Manini et al. [78] used a “Classification and regression trees” (CART) analysis to identify HGS cut points that best identify older adults with a slow walking speed, defined by three UGS cut-off points (<0.6 m/s; <0.8 m/s; and <1.0 m/s), and also used UGS as a continuous variable. A sensitivity analysis was performed on thirty-five sarcopenia candidate predictors (including HGS indices) that entered into the sex-stratified CART model and identified HGS/BMI (cut-off point of <1.46 kg/m^2^) in men and HGS/BW (cut-off point of <0.337) in women, using UGS, as the best predictors of mobility limitations (46% of men and 49% of women scored below the cut-off points). It must be emphasized that the SDOC CART model did not include HGS/WC as one of their sarcopenia candidate predictors. Manini’s subjects were much slimmer compared to our subjects (BMI ~26) (PBF 27% in men and 37% in women), with a much lower BW, hindering comparisons between studies [78]. 

Notwithstanding, individual mobility tests evaluate different mobility constructs, including gait, balance, transfer, or endurance [1]; a great deal of studies determine mobility limitations based on a single mobility test [78,79], making direct comparisons with our study challenging at best. This is especially true when mobility limitations are determined by walking short distances in a straight line (UGS), where many of the physiological systems needed for daily mobility challenges are not simulated [21]. Our finding that an additional 20% of men and 14% of women would have been considered mobility-intact if only a single test would have been performed is a compelling example of the shortcomings of performing a single mobility test for the identification of mobility-intact older adults. That said, performing multiple mobility tests can be time-consuming, requiring both staff training [80] as well as availability [12], and can potentially overburden older patients due to health risks or fatigue [24]. Therefore, if the mobility status of older adults with T2DM could be evaluated by an easy-to-measure HGS/WC, differentiating between mobility-intact and patients needing further mobility evaluation, the performance of multiple mobility and body composition tests may become redundant. Our finding that in older T2DM patients habitual physical activity was associated with an enhanced mobility status was shared by Fritschi et al., finding it to be a significant predictor of mobility [81]. Haskell et al., 1994 [82], observed that in inactive older adults, such as older T2DM patients, the performance of any amount of physical activity will result in relatively large improvements in mobility [82].

The main strength of this study is that this is the first study to evaluate relative strength using HGS and WC to identify mobility-intact older adults with T2DM.

Additionally, combining the different domains associated with each mobility test and giving them equal importance in the identification of mobility limitations increase the validity of identifying patients free of mobility limitations. This is especially true for adding the 2MWT, which has not been done before. Another strength of this research is that mobility tests’ cut-off points were in relation to the general population, as opposed to using percentiles of the studied T2DM population. Using the study’s population percentiles can possibly render a greater proportion of subjects as false negatives, thus falsely possessing an adequate mobility status.

This investigation has three main weaknesses: its cross-sectional design, its relatively low number of study participants, and the study of type 2 diabetes patients that possess specific medical, physical, and nutritional conditions, as specified in our eligibility criteria section. These limitations prevent us from declaring our HGS/WC cut-off points as determinants of intact mobility that can be applied to either larger or diverse ethnic diabetic populations. Additionally, as a control group of older adults without diabetes was not included, the study results can only be applied to older adults with type 2 diabetes, particularly type 2 diabetes patients that possess the medical, physical, and nutritional conditions specified in our eligibility criteria section.

## 5. Conclusions

In a cohort of primarily obese older men and women, the use of HGS relative to WC cut-off points was found to differentiate between mobility-intact patients and those at risk of mobility limitations, requiring further evaluation. Aside from identifying mobility limitations, the use of such measurements in a clinical setting can potentially identify patients with pre-sarcopenia via the use of established HGS cut-off points, along with identifying central obesity risk, both significant and prevalent health conditions associated with T2DM.

## Figures and Tables

**Figure 1 biomedicines-11-00352-f001:**
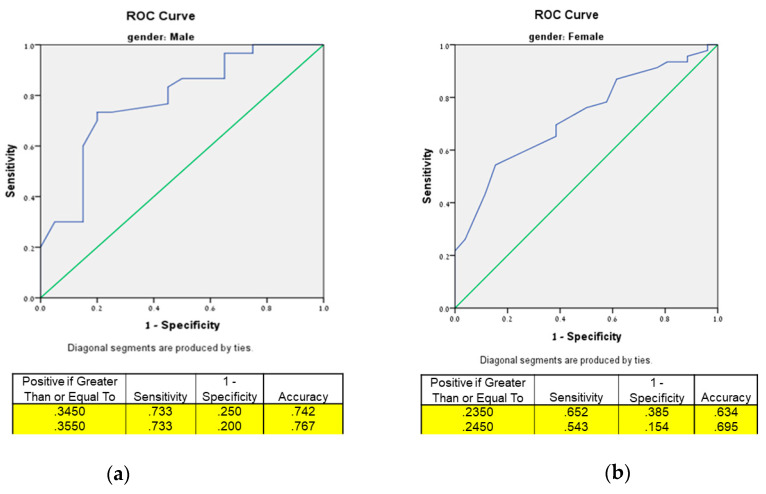
Receiver operating characteristic curve for identifying mobility-intact older T2DM patients using HGS/WC along with their corresponding optimal cut-off points. (**a**) Male, AUC = 0.781 (95% CI 0.649–0.913); (**b**) female, AUC = 0.725, (95% CI 0.609–0.842).

**Table 1 biomedicines-11-00352-t001:** Characteristics of study participants.

Variable	Total N = 122	Women N = 72	Men N = 50	*p*
Age (years)	70.3 (4.4)	70.3 (4.4)	70.1 (3.9)	0.82
Height (cm)	163 (8.8)	158 (5)	171 (6)	<0.001 *
Weight (kg)	85.4 (16.5)	79.9 (15)	93.2 (15.6)	<0.001 *
BMI (kg/m^2^)	32.2 (6.1)	32.3 (6.6)	31.9 (5.3)	0.68
BMI ≥ 30 (%) ^1^	63.9	64.0	63.8	0.68
WC (cm)	111.4 (14.7)	109.7 (15.4)	113.9 (13.4)	0.12
WC ≥ 102 cm men (%) ^2^ WC ≥ 88 cm women (%) ^2^	94.2	97.2	90.0	0.12
SMM (kg)	31.4 (12)	27.5 (10.3)	38.3 (11.5)	<0.001 *
BFM (kg)	34.9 (11)	35.8 (10.4)	35.1 (6.7)	<0.001 *
PBF Men ≥ 32.3; women ≥ 44.1 (%) ^3^	40.3 (7.5)	44.0 (5.6)	35.1 (6.8)	<0.001 *
Diabetes duration (years)	13.7 (9.3)	14.2 (8.7)	12.8 (10.2)	0.44
HbA1c (%)	7.4 (1.1)	7.4 (1)	7.56 (1.2)	0.43
Vitamin D (<25 ng/mL) (%)	31.1 (16.8)	37.5 (15.9) (*n* = 67)	40.5 (18.2) (*n* = 42)	0.98
Hypertension (%)	59.9	62.0	58.4	0.68
Diabetes complications ^4^ (%)	39.4	31.9	42.0	0.25
Polypharmacy (≥8 drugs) (%)	32.1	33.3	30.6 (*n* = 49)	0.75
Physical activitys ^5^ (%)	50.8	51.4	50.0	0.88
** *HGS Indices* **				
HGS (absolute) (kg)	31.3 (9.4)	24.9 (4.6)	40.44 (6.6)	<0.001 *
HGS/BW (kg/kg)	0.37 (0.1)	0.32 (0.07)	0.44 (0.1)	<0.001 *
HGS/height (kg/m)	19.03 (5)	15.82 (2.8)	23.64 (3.6)	<0.001 *
HGS/BMI (kg/kg/m^2^)	1.01 (0.3)	0.80 (0.2)	1.30 (0.31)	<0.001 *
HGS/WC (kg/cm)	0.28 (0.1)	0.23 (0.05)	0.36 (0.08)	<0.001 *
HGS/SMM (kg/kg)	1.04 (0.29)	0.99 (0.28)	1.12 (0.28)	<0.001 *
HGS/BFM (kg/kg)	1.0 (0.5)	0.76 (26)	1.36 (0.55)	<0.001 *
HGS/PBF (kg/%)	0.84 (0.4)	0.58 (0.13)	1.21 (0.38)	<0.001 *

Notes: Variables are presented as mean ± SD/% of the population. * Statistically significant difference. ^1^ CDC, Division of Nutrition, Physical Activity, and Obesity, National Center for Chronic Disease Prevention and Health Promotion. 2022. ^2^ Grundy SM. Diagnosis and management of the metabolic syndrome: an American Heart Association/National Heart, Lung, and Blood Institute Scientific Statement. Circulation 2005. ^3^ Heo M. Percentage of body fat cut-offs by sex, age, and race-ethnicity in the US adult population from NHANES 1999-2004. Am J Clin Nutr. 2012. ^4^ Diabetes’s complications: neuropathy, retinopathy, chronic kidney disease, ischemic heart disease. ^5^ Performing ≤ 2 days a week of any leisure aerobic physical activity. Abbreviations: BFM, body fat mass; BMI, body mass index; BW, body weight; H, height; HGS, hand grip strength; HbA1c, hemoglobin A1c; PBF, percent body fat; SMM, skeletal muscle mass; and WC, waist circumference.

**Table 2 biomedicines-11-00352-t002:** Mobility characteristics of the participants in the cohort.

Test	Total N = 122 (%)	Women N = 72	Men N = 50	*p*
UGS (m/sec)	1.11 (0.2)	1.08 (0.2)	1.14 (0.2)	0.06
TUG (sec)	10.68 (0.2)	10.97 (2.5)	10.25 (1.8)	0.03 *
2MWT (m)	162.9 (31.3)	154.2 (28.2)	175.6 (31.5)	<0.001 *
Pass UGS (%) ^1^	83.6	87.5	78.0	0.16
Pass TUG (%) ^2^	79.5	77.8	82.0	0.57
Pass 2MWT (%) ^3^	73.7	75.0	71.4	0.66
Pass all tests (%)	63.3	63.9	60.0	0.66
Fail 1 test only (%)	20.4	20.9	20.0	0.878
Fail > 1 test	16.4	13.9	20.0	0.977

Notes: * Statistically significant difference. ^1^ A score equal or above the 25th percentile of the "NIH 4-Meter Walk gait speed test" (stratified by age and gender) [62]. ^2^ A time of <12 s [18]. ^3^A score equal or above the 25th percentile of the "NIH Toolbox 2 Minute Walk Endurance Test" (stratified by age and gender) [62]. Abbreviations: UGS, usual gait speed; TUG, timed up and go; and 2MWT, 2-minute walk test.

**Table 3 biomedicines-11-00352-t003:** Area under the receiving operator curve (AUC) and associated data that best predicted passing each and all three mobility tests (being “mobility-intact”) using HGS/WC in a cohort of older T2DM patients.

Test	AUC (95% CI)	*p*-Value	Cut-Off	Sensitivity %	Specificity %	Accuracy
**Men**						
UGS	0.758 (0.579, 0.936)	0.01 *	0.305	0.821	0.636	0.728
TUG	0.827 (0.704, 0.950)	0.002 *	0.355	0.634	1.0	0.817
2MWT	0.779 (0.644, 0.913)	0.003 *	0.335	0.686	0.857	0.771
Pass all	0.781 (0.649, 0.913)	0.01 *	0.355	0.733	0.800	0.767
**Women**						
UGS	0.631 (0.402, 0.861)	0.2	0.180	0.825	0.556	0.690
TUG	0.725 (0.602, 0.848)	0.006 *	0.235	0.500	0.933	0.719
2MWT	0.751 (0.623, 0.880)	0.002 *	0.215	0.685	0.722	0.704
Pass all	0.725 (0.609, 0.842)	0.002 *	0.245	0.543	0.846	0.695

Notes: * Statistically significant difference. Abbreviations: AUC, area under the curve; CI, confidence interval; UGS, usual gait speed; TUG, timed up and go; and 2MWT, 2-minute walk test.

**Table 4 biomedicines-11-00352-t004:** Odds ratio (OR) and 95% CI for passing individual and all three mobility tests using HGS/WC cut-off points.

Test	Gender	N	OR	95% CI
UGS	Men	50	8.0	(1.3–34.9)
	Women	72	0.4	(0.37–4.3)
TUG	Men	49	*	*
	Women	72	15	(1.8–121.4)
2MWT	Men	49	13.1	(2.5–68.7)
	Women	72	5.6	(1.7–18.4)
Pass all	Men	50	8.2	(2.2–30.1)
	Women	72	3.0	(1.1–8.1)

Notes: * could not be computed. Abbreviations: OR, odds ratio; CI, confidence interval; UGS, usual gait speed; TUG, timed up and go; and 2MWT, 2-minute walk test.

## Data Availability

Data are available from the authors upon reasonable request and with the permission of the CEV-65 investigators.

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
