# Peer review of "Hand Grip Strength Relative to Waist Circumference as a Means to Identify Men and Women Possessing Intact Mobility in a Cohort of Older Adults with Type 2 Diabetes"

_biomedicines, 2023, doi:10.3390/biomedicines11020352_

Round 1

Reviewer 1 Report

The manuscript entitled: ``Can relative hand-grip strength identify mobility-intact in a cohort of older adults with type 2 diabetes`` is an original study. The authors assessed the hand grip index to evaluate mobility in older adults with T2DM (defined as older than 65 years). Even it is well written, there are some important issues in this study. The theme could be important for future clinical application.

Major revision

The title seems not appropriate. The title is a question? Please reformulate according with study results. Actually, the authors studied gender differences in these patients. All tables compare the parameters evaluated in women versus men.

Why do you not use a control group (patients >65 years without diabetes)?

Study limitations are poor.

Minor revision

Page 2 line 48: What means TUG? Please explain the all acronyms. Please put an appropriate legend for table 1 (LBM, BFM, etc). Each table must have a legend with all acronyms.

Page 10 lines 349: ``…to identify mobility intact older adults with T2DM and patients.`` Please reformulate.

There are some language errors.

Author Response

Major revision

Comment 1. The title seems not appropriate. The title is a question? Please reformulate according with study results. Actually, the authors studied gender differences in these patients. All tables compare the parameters evaluated in women versus men.

Thank you very much for comment! The title of the manuscript has been changed and now reads: “Hand-grip strength relative to waist circumference as a means to identify men and women possessing intact mobility in a cohort of older adults with type 2 diabetes". (line 1-6)

Comment 2. Why do you not use a control group (patients >65 years without diabetes)?

Thank you very much for your comment!

This manuscript is a sub study based upon analysis of two investigations studying changes in physical abilities between three different groups of older adults with type 2 diabetes receiving three interventions: a. medication (empagliflozin), b. vegetarian diet (with added protein), c. home-based exercise with minimal supervision. This is the main reason for not including a control group of non-diabetic older adults.

Comment 3. Study limitations are poor

The study limitations section was modified and now reads:

This investigation has three main weaknesses: its cross-sectional design, its relatively low number of study participants, and the study of type 2 diabetes patients that possess specific medical, physical, and nutritional conditions as specified in our eligibility criteria section. These limitations prevent us from declaring our HGS/WC cut-off points as determinants of intact mobility that can be applied to either larger or diverse ethnic diabetic populations. Also, as a control group of older adults without diabetes was not included, the study results can only be applied to older adults with type 2 diabetes. In particular, type 2 diabetes patients that possess the medical, physical, and nutritional conditions specified in our eligibility criteria section. (line 369-377)

Minor revision

Comment 1. Page 2 line 48: What means TUG? Please explain all acronyms.

TUG mean timed up and go. All acronyms including TUG were expanded (line 52).

Comment 2. Please put an appropriate legend for table 1 (LBM, BFM, etc). Each table must have a legend with all acronyms

The legend for table 1 was revised to contain all appropriate acronyms. 

Comment 3. Page 10 lines 349: ``…to identify mobility intact older adults with T2DM and patients.`` Please reformulate. There are some language errors (line 351).

The sentence was changed  as the words “and patients” have been omitted (line 360).

Reviewer 2 Report

Kis et al., studied that whether the hand-grip strength index can be used to predict mobility limitations in type-2 diabetic population. The study is well- designed and the results are explained well. 

There are few suggestions:

1. Line 140: "Cut-off point for passing the TUG test for both genders was considered a time of >12 seconds".

According to the reference (reference 18), >12 seconds indicates an increased risk of fall. The results (table 2) shows the average TUG value of the cohort is <12. Kindly explain.

2. Line 148, Line 64 - The abbreviation can be expanded for TUG and UGS.

3. Line 141: heading 2.4.3 : It should be mentioned as Two-minute walk test instead of Minute walk test. 

Author Response

Comment 1. Line 140: "Cut-off point for passing the TUG test for both genders was considered a time of >12 seconds". According to the reference (reference 18), >12 seconds indicates an increased risk of fall. The results (table 2) show the average TUG value of the cohort is <12. Kindly explain.

Thank you for this important comment!. A correction was made, and the sentence was changed to : “Cut-off point for passing the TUG test for both genders was considered a time of <12 seconds! (line 145). 

Comment 2. Line 148, Line 64 - The abbreviation can be expanded for TUG and UGS.

Abbreviations were expanded for T2DM (line 48) and BMI (line66). TUG was expanded (line 54) and UGS was already expanded on line 141.

Comment 3. Line 141: heading 2.4.3 : It should be mentioned as Two-minute walk test instead of Minute walk test.

Heading 2.4.3 was changed to “Two-minute walk test”. (line 148)

Round 2

Reviewer 1 Report

Thank you for responding to my comments/suggestions.